Effect of blood-flow restricted vs heavy-load resistance training on strength, power, and speed for healthy volunteers: a systematic review and meta-analysis

Wang Tao 1
Liu Yutong 2
Wang Xiaolin 1
http://orcid.org/0000-0002-5180-7862 Amri Saidon 1
Kamalden Tengku Fadilah 3
http://orcid.org/0000-0002-6013-2385 Gao Zhendong 1
http://orcid.org/0000-0001-7373-2139 Ng Yee Guan 4 shah86zam@upm.edu.my
1 Faculty of Educational Studies, Universiti Putra Malaysia , Serdang, Selangor , Malaysia
2 Department of Sports Teaching and Research, Lanzhou University , Lanzhou, Gansu , China
3 National Sports Complex , Kuala Lumpur, Selangor , Malaysia
4 Faculty of Medicine and Health Sciences, Universiti Putra Malaysia , Serdang, Selangor , Malaysia
Centner Christoph
Electronic publication date: 2025 Mar 18
Publication date: 2025
Volume: 13
Electronic Location ID: e19110
Received 2024 Oct 3; Accepted 2025 Feb 13
Copyright: © 2025 Wang et al.
Copyright year: 2025
Copyright holder: Wang et al.
License: This is an open access article distributed under the terms of the Creative Commons Attribution License, which permits unrestricted use, distribution, reproduction and adaptation in any medium and for any purpose provided that it is properly attributed. For attribution, the original author(s), title, publication source (PeerJ) and either DOI or URL of the article must be cited.
License URL: https://creativecommons.org/licenses/by/4.0/

Keywords: Blood flow restriction training, High-load resistance training, Muscle strength, Power, Speed

Funding: The authors received no funding for this work.

==============================
Background

Low-load blood flow restriction (LL-BFR) training has been shown to enhance muscle strength, power, and speed, but its effectiveness compared to traditional high-load resistance (HLR) training remains unclear. This meta-analysis aimed to compare the effects of LL-BFR and HLR training on muscle strength, power, and speed.

Methodology

Studies were identified by searching the SCOPUS, SPORTDiscus, PubMed, Web of Science, and CNKI databases up to May 13, 2024, using the following inclusion criteria: (a) healthy population; (b) comparison of LL-BFR vs HLR training; (c) pre- and post-training assessment of muscle strength (dynamic, isometric, and isokinetic), muscle power, jump, or speed performance; (d) PEDro scale score ≥4. The methodological quality of the included studies was assessed using the PEDro tool and the Grading of Recommendations Assessment, Development, and Evaluation (GRADE) approach, with meta-analyses conducted using the R program.

Results

A total of 41 studies, involving 853 subjects, were included in the meta-analysis. Based on the PEDro scores and GRADE assessment, the overall quality of the included studies was assessed as moderate. LL-BFR training showed a slightly smaller effect on maximal strength compared to HLR training (ES = −0.19, 95% CI [−0.31 to −0.06], p < 0.01). There were no significant differences between LL-BFR and HLR training for muscle power (ES = −0.04, 95% CI [−0.33 to 0.24], p > 0.05), jump performance (ES = −0.08, 95% CI [−0.30 to 0.15], p > 0.05), and speed (ES = −0.28, 95% CI [−0.71 to 0.15], p > 0.05). Additionally, individual characteristics (i.e., age, gender, and training status) and training parameters (i.e., training duration, frequency, cuff pressure, and cuff width) did not significantly moderate the training effect.

Conclusions

LL-BFR training showed slightly less improvement in maximal strength compared to HLR training but demonstrated comparable effects on muscle power, jump performance, and speed in healthy individuals in healthy individuals. These findings suggest that LL-BFR may be a practical and effective alternative for individuals seeking performance improvements with lower training loads.

Introduction

Traditional high-load resistance training (HLR; i.e., > 65% of one repetition maximum (1RM)) is widely regarded as an effective strategy for enhancing maximal strength, muscle power, and speed (Kraemer & Ratamess, 2004; Marshall et al., 2021). The American College of Sports Medicine (ACSM) also recommends high-load training using external loads of 60–90% 1RM to improve maximal muscle strength and power (Garber et al., 2011). However, high-load training may not be suitable for older adults, untrained individuals, and short-season athletes, as it may increase the risk of injury, discomfort, and negatively affect athletic performance during competitions (Liu & Latham, 2010; Slysz, Stultz & Burr, 2016). This is because high-load resistance training can increase the risk of injury and discomfort, thereby affecting the continuity and effectiveness of the training. Consequently, there has been a growing interest in exploring low-load alternatives to high-load exercise.

Low-load resistance training typically has minor or no positive effects on strength, power, and speed compared to traditional HLR training (Schoenfeld et al., 2016; Suetta et al., 2004; McBride et al., 2002). However, in recent years, low-load blood flow restriction resistance training (LL-BFR) has gained increasing attention as a promising alternative (Burton, 2022; Hughes et al., 2017). Low-load blood flow restriction training (LL-BFR), which combines low external loads (20–30% 1RM) with partial blood flow restriction, induces metabolic stress and muscle fatigue, leading to adaptations in muscle strength, hypertrophy, and power that are comparable to those from high-load training (Jessee et al., 2018; Pearson & Hussain, 2015; Lixandrão et al., 2018). Therefore, LL-BFR training may provide a viable alternative to HLR training.

LL-BFR training has been shown to significantly improve muscle strength, power, and speed, with promising effects not only in healthy individuals and athletes (Centner et al., 2019; Vergara et al., 2024; Wortman et al., 2021), but also in certain pathological populations (Ahmed, Mustafaoglu & Erhan, 2024; Nitzsche et al., 2021; Wen et al., 2023). Previous studies have found that there were comparable effects of LL-BFR training and HLR training on muscle strength, muscle power, and speed in healthy individuals (Castilla-López & Romero-Franco, 2023; Wang et al., 2022; Laurentino et al., 2012). However, some studies indicated that LL-BFR training was less effective compared to HLR training in terms of training gains (Clark et al., 2011; de Lemos Muller, Ramis & Ribeiro, 2019; Ramis et al., 2020). Besides the risk of Type II error (i.e., failing to detect a true effect due to factors like insufficient sample size or high variability), these discrepancies may be attributed to differences in training variables (e.g., training duration, training frequency, cuff pressure, and cuff width) and subject characteristics (Lixandrão et al., 2018; Xiaolin et al., 2023; Fabero-Garrido et al., 2022). Differences in these factors can lead to different physiological adaptations, ultimately impacting the effectiveness of the training outcomes. Therefore, it is necessary to conduct a meta-analysis to compare the effects of LL-BFR and HLR training on muscle strength, power, and speed in healthy people comprehensively. However, existing meta-analyses only compared the effects of LL-BFR and HLR training on muscle strength (Lixandrão et al., 2018; Grønfeldt et al., 2020), lacking systematic reviews and meta-analyses that thoroughly compared their impacts on muscle strength, power, and speed.

Therefore, the aim of this study was to conduct a systematic review and meta-analysis to compare the effects of LL-BFR training and HLR training on muscle strength, muscle power, and speed in healthy people, thereby providing scientific evidence for related training practices. Additionally, this study aims to explore the potential moderating effects of individual characteristics (i.e., age, gender, and athletic level) and training variables (i.e., training duration, frequency, cuff pressure, and cuff width) on the responses in muscle strength, muscle power, and speed.

Methods

Registration and literature search

This meta-analytical review was performed in accordance with the PRISMA statement guidelines for Systematic Reviews and Meta-Analyses (Page et al., 2021) (Prospero registration number: CRD42024536163).

The articles published by May 13, 2024, were identified through the SCOPUS, SPORTDiscus, PubMed, Web of Science, and CNKI databases. The search strategy combined the following keywords: “blood flow restriction”, “vascular occlusion”, “KAATSU”, “strength training”, “resistance training”, “high intensity training”, “weight training”, “high load training”, to ensure comprehensive coverage of relevant literature. The full search string is provided in Supplemental Material 1. After removing duplicates, the screening process was conducted in three phases: titles, abstracts, and full texts (Fig. 1). To further ensure the inclusion of all relevant studies, reference lists from included studies, previous reviews, and the lead authors’ personal libraries were also manually examined. To enhance methodological rigor, two assessors (Y.L. and Z.G.) independently retrieved articles and extracted data. Discrepancies were resolved by a third author (T.W.).

Figure 1 PRISMA flow diagram.

Eligibility criteria

Articles were eligible for inclusion based on the following criteria: (a) healthy population; (b) pre- and post-training assessment of muscle strength (i.e., dynamic, isometric, and isokinetic), muscle power, jump, or sprint speed performance; (c) comparisons of HLR (>65% 1RM) vs LL-BFR (<50% 1RM); (d) The literature quality assessment (Physiotherapy Evidence Database, PEDro) scale scores ≥4. The eligibility criteria were established to ensure a homogeneous study population, comparable interventions, and reliable meta-analysis outcomes by excluding low-quality or incomplete studies.

Methodological quality assessment and risk of bias

The methodological quality of the included studies was assessed with the PEDro scale (Verhagen et al., 1998). This scale consists of 11 items (e.g., randomization, blinding, and outcome measures), with a maximum total score of 10 points (item 1 is not rated). In accordance with prior research (Stojanović et al., 2017; Wang et al., 2024), studies scoring below 4 on the PEDro scale were considered low quality and excluded to ensure the reliability of the meta-analysis. Therefore, only studies with a PEDro score of four or higher were included in this meta-analysis. Additionally, the GRADE approach was used to assess the overall quality of the evidence (Guyatt et al., 2021). Two reviewers (X.W. and S.A.) conducted the methodological quality assessment, resolving any discrepancies through consensus with the third author (T.W.). Additionally, the potential for bias was evaluated through visual inspection of funnel plots and Egger’s test, ensuring transparency and minimizing potential bias in the analysis.

Data extraction

Primary outcomes extracted from each study included muscle strength (isometric and isokinetic), power (e.g., countermovement jump, squat jump, depth jump, and long jump), and sprint speed (10–30 m sprint). Secondary outcomes focused on the moderating effects of participant characteristics and training variables on the primary outcomes. Participant characteristics included age (years), gender, and training status (resistance-trained or non-resistance-trained). Training protocol details included training duration (weeks), frequency (days/week), exercise load (% 1RM), exercise mode, cuff pressure, and cuff width. For studies with multiple assessment time points (e.g., at 3 weeks, 6 weeks, and the final week of training), the data from the final measurement at the end of the training period were included in the analysis to ensure consistency across studies and avoid variability from mixed time points, thereby minimizing methodological heterogeneity. Data extracted are available in Table 1 and Supplementary Material 2.

Table 1 LL-BFR vs HLR training and changes in muscle strength, power, and speed.

Study	Subjects	Protocol	Duration/frequency	N, %F	Exercise mode	Outcomes (percentage increase)	
Bradley et al. (2023)	Untrained adults
30.2 ± 7.7 years	LL-BFR (30–40% 1RM)
HLR (60–80% 1RM)	4 weeks
2 days/week	9/11
NG	Rowing, deadlift	1RM Deadlift: BFR 20%, HLR 18.5%
Peak power: BFR 12.5%, HLR 9.8%	
Castilla-López & Romero-Franco (2023)	Male soccer players
19.2 ± 1.7 years	LL-BFR (20% 1RM)
HLR (70% 1RM)	6 weeks
2 days/week	9/9
0%	Back squat, deadlift, and hip thrust	CMJ: BFR 5.4%, HLR 5.4%
MVC vertical force: 4.5%, HLR 0.5%
Peak power: BFR –2.4%, HLR –11.9%
30-m Sprint: BFR 2.3%, HLR 4.4%	
Centner et al. (2019)	Untrained adults
27.9 ± 5.1 years	LL-BFR (20–35% 1RM)
HLR (70–85% 1RM)	14 weeks
3 days/week	11/14
0%	Dynamic standing and sitting calf raises	MVC Plantar flexors: BFR 9.8%, HLR 13.5%	
Centner et al. (2023)	Untrained adults
27.6 ± 4.3 years	LL-BFR (20–35% 1RM)
HLR (70–80% 1RM)	14 weeks
3 days/week	14/15
0%	Sitting and standing calf raises	MVC Plantar flexors: BFR 43.5%, HLR 43.6%	
Clark et al. (2011)	Untrained adults
24 ± 1.6 years	LL-BFR (30% 1RM)
HLR (80% 1RM)	4 weeks
3 days/week	9/7
12%	Leg extension	MVC Leg extension: BFR 6.4%, HLR 11.9%	
Cook et al. (2018)	Untrained adults
19.8 ± 1.3 years	LL-BFR (30% 1RM)
HLR (70% 1RM)	6 weeks
3 days/week	6/6
50%	Leg extension and press	1RM Knee extension: BFR 12.9%, HLR 34.1%	
Cook & Cleary (2019)	Older adults
76.3 ± 7.8 years	LL-BFR (30% 1RM)
HLR (70% 1RM)	12 weeks
2 days/week	10/11
57%	Knee extension and flexion	MVC Leg extension: BFR 10.5%, HLR 21.4%
MVC Leg flexion: BFR 9.6%, HLR 13.4%	
Davids et al. (2021)	Trained men
24.3 ± 3.1 years	LL-BFR (30–40% 1RM)
HLR (75–80% 1RM)	9 weeks
3 days/week	11/10
NG	Barbell squats, leg presses, leg extensions, and Bulgarian split squats	1RM Leg extension: BFR 4.3%, HLR 8.6%
CMJ: BFR 3.7%, HLR 4.8%
CMJ power: BFR 0.9%, HLR 1.8%
Squat jump: BFR −0.3%, HLR 7.0%
Squat jump power: BFR 0.9%, HLR 1.8%	
Early et al. (2020)	Untrained adults
23 ± 4 years	LL-BFR (30% 1RM)
HLR (60% 1RM)	8 weeks
2 days/week	11/10
65%	Leg extension, leg curl, and heel raise	1RM Leg curl: BFR 23.8%, HLR 32.1%
1RM Leg extension: BFR 19.6%, HLR 20.9%
1RM Heel raise: BFR 29.4%, HLR 29.9%	
Guang (2021)	Male basketball players
20.6 ± 1.0 years	LL-BFR (30% 1RM)
HLR (85% 1RM)	6 weeks
3 days/week	8/8
0%	Back squat, depth jump, and sprint	1RM Back squat: BFR 9.9%, HLR 11.2%
CMJ: BFR 6.5%, HLR 4.9%
Sprint: BFR 3.3%, HLR 2.2%	
Horiuchi, Stoner & Poles (2023)	Untrained men
22.0 ± 2.0 years	LL-BFR (30% 1RM)
HLR (75% 1RM)	6 weeks
4 days/week	12/12
0%	Knee extensions and leg presses	1RM Knee extension: BFR 6.6%, HLR 14.7%
1RM Leg press: BFR 12.1%, HLR 14.0	
Jeon (2022)	Untrained women
47.8 ± 5.2 years	LL-BFR (30% 1RM)
HLR (65–70% 1RM)	8 weeks
3 days/week	9/7
100%	Leg extension and curl	1RM Knee extension: BFR 6.6%, HLR 13.5%
1RM Knee flexion: BFR 22.8%, HLR 24.6%	
Jones et al. (2023)	Untrained adults
24.3 ± 2.6 years	LL-BFR (30% 1RM)
HLR (80% 1RM)	6 weeks
2 days/week	20/20
47%	Leg extension and curl	Vertical power: BFR 1.5%, HLR 2.3%
1 RM Hamstrings force: BFR 9.4%, HLR 12.5%	
Karabulut et al. (2010)	Older men
56.8 ± 0.6 years	LL-BFR (20% 1RM)
HLR (80% 1RM)	6 weeks
3 days/week	13/13
0%	Leg extension and press	1RM Leg extension: BFR 18.1%, HLR 30.8%
1RM Leg press: BFR18.7%, HLR 18.4%	
Kim et al. (2009)	Untrained men
23.5 ± 1.2 years	LL-BFR (20% 1RM)
HLR (80% 1RM)	3 weeks
3 days/week	10/10
0%	Leg press, knee flexion, and knee extension	1RM Leg press: BFR 11.1%, HLR 15.0%
1RM Knee flexion: BFR 3.6%, HLR 12.4
1RM Knee extension: BFR 7.1%, HLR 18.1%	
Kriley (2014)	Male football players
20.3 ± 1.1 years	LL-BFR (20% 1RM)
HLR (65–90% 1RM)	7 weeks
2 days/week	16/15
0%	Back squat	1RM Squat: BFR 3.4%, HLR 7%
Vertical jump: BFR 1.9%, HLR 4.4%	
Laurentino et al. (2012)	Untrained male students
21.8 ± 5.4 years	LL-BFR (20% 1RM)
HLR (80% 1RM)	8 weeks
2 days/week	10/9
0%	Leg extension	1RM Knee extension: BFR 39.7%, HLR 35.6%	
Letieri et al. (2018)	Older women
68.8 ± 5.1 years	LL-BFR (20–30% 1RM)
HLR (70–80% 1RM)	16 weeks
3 days/week	11/10
100%	Squat, leg press, leg curl, knee extension	Peak torque leg extension: BFR 17.8%, HLR 28.4%
Peak torque Leg flexion: BFR 23%, HLR 19.1%	
Li (2020)	Male handball players
24 ± 4.3 years	LL-BFR (30–40% 1RM)
HLR (60–80% 1RM)	8 weeks
3 days/week	9/9
0%	Squat, deadlift, lunge, and sprint	1RM Squat: BFR 6.6%, HLR 4.4%
Vertical jump: BFR 11.2%, HLR 13.4%
30m Sprint: BFR 4.4%, HLR 2.7%	
Lixandrão et al. (2015)	Untrained men
26.9 ± 8.4 years	LL-BFR (20–40% 1RM)
HLR (80% 1RM)	12 weeks
2 days/week	42/9
0%	Unilateral knee extension	1RM Knee extension: BFR1 10.3%, BFR2 13.2%, BFR3 12.2%, BFR4 12.6%, HLR21.6%	
Luebbers et al. (2019)	Adolescent weightlifters
15.9 ± 1.2 years	LL-BFR (20% 1RM)
HLR (65–80% 1RM)	6 weeks
2 days/week	8/9
12%	Back squat, clean & jerk, and deadlift	1RM Parallel back squat: BFR 16.5%, HLR 7%	
Luebbers et al. (2014)	Male Football players
20.3 ± 1.1 years	LL-BFR (20% 1RM)
HLR (65–90% 1RM)	7 weeks
2 days/week	16/15
0%	Squat, Glute-Ham raise, and DB lunge	1RM Squat: BFR 3.4%, HLR 6.9%	
Martín-Hernández et al. (2013)	Untrained students
20.5 ± 1.8 years	LL-BFR (20% 1RM)
HLR (85% 1RM)	5 weeks
2 days/week	10/11
0%	Leg extension	1RM Knee extension: BFR 4.7%, HLR 6.5%
1RM Knee flexion: BFR −1.3%, HLR −2.6%	
May et al. (2022)	Untrained men
24.1 ± 3.9 years	LL-BFR (30% 1RM)
HLR (70% 1RM)	7 weeks
3 days/week	8/9
0%	Knee flexion and extension	1RM Knee extension: BFR 17.8%, HLR 18.8%
1RM Knee flexion: BFR 10.5%, HLR 14.3%	
Mendonca et al. (2021)	Untrained adults
22.1 ± 3.1 years	LL-BFR (20% 1RM)
HLR (75% 1RM)	4 weeks
2 days/week	15/15
47%	Plantar flexion and dorsiflexion	MVC Plantarflexion: BFR 16.3% HLR 18.1%
Rate of torque development: BFR 17.4%, HLR 18.3%	
de Lemos Muller, Ramis & Ribeiro (2019)	Untrained men
23 ± 2.7 years	LL-BFR (30% 1RM)
HLR (80% 1RM)	8 weeks
3 days/week	13/13
0%	Leg extension	1RM Knee extension: BFR 30.8%, HLR 45.7%	
Ramis et al. (2020)	Untrained men
24 ± 2.7 years	LL-BFR (30% 1RM)
HLR (70% 1RM)	8 weeks
3 days/week	15/13
0%	Knee extension	Peak torque knee extension: BFR 10.5%, HLR 21.6%	
Reece et al. (2023)	Untrained adults
21.8 ± 3.1 years	LL-BFR (30% 1RM)
HLR (80% 1RM)	6 weeks
3 days/week	15/15
53%	Leg extension	1RM Leg extension: BFR1 16.5%, HLR 49.0%; BFR2 17.5%, HLR 28.6%	
Seynnes et al. (2022)	Untrained men
27.4 ± 2.6 years	LL-BFR (20–35% 1RM)
HLR (70–85% 1RM)	14 weeks
3 days/week	15/14
0%	Leg press, knee extensions, standing calf raises, and seated calf raises	1RM Leg press: BFR 33.5%, HLR 37.5%	
Shao (2023)	Male basketball players
17.3 ± 0.5 years	LL-BFR (20–30% 1RM)
HLR (60–85% 1RM)	6 weeks
3 days/week	8/8
0%	Squat, deadlift, heel raise, and lunge	1RM Squat: BFR 10.1%, HLR 12.4%
CMJ: BFR 3.1%, HLR 2.3%
Horizontal jump: BFR 3.1%, HLR 2.1%
Sprint: BFR 2.6%, HLR 2.3%	
Silva et al. (2015)	Older women
62.2 ± 4.5 years	LL-BFR (30% 1RM)
HLR (80% 1RM)	12 weeks
2 days/week	5/5
100%	Bicycle	1RM Leg extension: BFR 11.4%, HLR 35.3%	
Sousa et al. (2017)	Untrained adults
22.2 ± 4.4 years	LL-BFR (30% 1RM)
HLR (80% 1RM)	6 weeks
2 days/week	10/11
43%	Leg extension	Peak torque Knee extension: BFR 21.4%, HLR 42.3%	
Tang & Qu (2022)	Male cyclists
23 ± 1.8 years	LL-BFR (40% 1RM)
HLR (75% 1RM)	4 weeks
2 days/week	6/6
0%	Half squat, deadlift, hip thrust, leg press	1RM Half squat: BFR 7.6%, HLR 4.1%
Sprint power: BFR 2.9%, HLR 1.6%	
Thiebaud et al. (2013)	Older women
61 ± 5 years	LL-BFR (10–30% 1RM)
HLR (70–90% 1RM)	8 weeks
3 days/week	6/8
100%	Knee extension & flexion, hip flexion & extension	1RM Leg press: BFR 7.7%, HLR 13.7%	
Vechin et al. (2015)	Older adults
64 ± 3.8 years	LL-BFR (20–30% 1RM)
HLR (70–80% 1RM)	12 weeks
2 days/week	8/8
39%	Leg press	1RM Leg press: BFR 15.6%, HLR 49.5%	
Wang et al. (2019)	Male handball players
24 ± 3.8 years	LL-BFR (30% 1RM)
HLR (70% 1RM)	8 weeks
3 days/week	9/9
0%	Deadlift, and back squat	CMJ: BFR 5.9%, HLR 9.5%
Squat jump: BFR 17.4%, HLR 22.4%
Peak power: BFR 3.5%, HLR 10.7%
30m Sprint: BFR 3.7%, HLR 3.4%	
Wang et al. (2022)	Male volleyball players
20.7 ± 0.5 years	LL-BFR (30% 1RM)
HLR (70% 1RM)	8 weeks
3 days/week	6/6
0%	Half squat	1RM Half squat: BFR 10%, HLR 17.3%
Horizontal jump: BFR 0.3%, HLR 1.2%
Squat jump: BFR 3.3%, HLR 5.8%
Peak power SJ: BFR 7.1%, HLR 10.2%
CMJ: BFR 1.1%, HLR 3.6%
Peak power CMJ: BFR 3.9%, HLR 5.2%	
Wang et al. (2023)	Male swimmers
19.7 ± 1.6 years	LL-BFR (30% 1RM)
HLR (70% 1RM)	4 weeks
3 days/week	8/8
0%	Back squat	1RM Back squat: BFR 14.7%, HLR 6.7%	
Xie (2023)	Male wrestlers
20.1 ± 0.4 years	LL-BFR (30% 1RM)
HLR (75% 1RM)	6 weeks
3 days/week	12/12
0%	Half squat, and lunge	Peak power Bicycle: BFR 6.3%, HLR 2.9%
Horizontal jump: BFR 7.9%, HLR 4.6%
30m Sprint: BFR 7.6%, HLR 3%
CMJ: BFR 8.1%, HLR 4.7%
1RM Back squat: BFR 2.1%, HLR 3.5%	
Yang et al. (2022)	Trampoline gymnasts
13.9 ± 0.4 years	LL-BFR (25–30% 1RM)
HLR (60–85% 1RM)	10 weeks
2 days/week	7/8
53%	Back squat, and jump	Squat jump: BFR 21.7%, HLR 17.1%
CMJ: BFR 19.6%, HLR 22.2%
Depth jump: BFR 10.4%, HLR 23.3%	
Yasuda et al. (2016)	Older women
71 ± 6.6 years	LL-BFR (35–45% 1RM)
HLR (70–90% 1RM)	12 weeks
2 days/week	10/10
100%	Bilateral squat, and knee extension	MVC Leg extension: BFR 13.7%, HLR 9.8%	
Zhang, Ye & Wu (2022)	Untrained women
23.3 ± 1 years	LL-BFR (20% 1RM)
HLR (75% 1RM)	5 weeks
2 days/week	8/8
0%	Back squat	Squat jump: BFR 1.7%, HLR 5.3%
1RM Back squat: BFR 2.2%, HLR 6.3%	
Note:

CMJ, countermovement jump; LL-BFR, low load blood flow restriction training; HLR, high load resistance training (traditional training); MVC, maximum voluntary contraction; N, sample size, LL-BFR/HLR; %F, percentage of females.

Statistical analyses

All meta-analyses were conducted by R packages (R version 4.3.0 with R Studio version 2024.04.1+748). The metacont() function from the meta package was used for conducting the meta-analyses, while subgroup analyses were performed using the update() function. Bias assessment was carried out using the metabias() function from the metafor package. Effect sizes (standardized mean difference, SMD) were calculated using the changes in means and standard deviations from pre- to post-intervention due to baseline differences in some included studies (Higgins, Li & Deeks, 2019). The change in standard deviation ( SDchange) was determined using the following equation:

SDchange=(SDpre2/Npre)+(SDpost2/Npost).

The magnitude of effect size was interpreted using the following scale: <0.2 (trivial), 0.2‒0.5 (small), 0.5‒0.8 (moderate), and >0.8 (large) (Cohen, 2013). Methodological heterogeneity, including differences in study design, participant characteristics, and intervention protocols, was systematically assessed and accounted for through subgroup analyses. Statistical heterogeneity was assessed by evaluating the variation in effect sizes across studies, the overlap of confidence intervals, as well as quantitatively using the I2 statistic. I2 values <25% indicating low heterogeneity, 25–75% indicating moderate heterogeneity, and >75% indicating high heterogeneity (Higgins et al., 2003). Considering measurement variability and heterogeneity among the included studies, a random effects model was applied.

A total of four meta-analyses were conducted. The analyses examined the impact of LL-BFR vs HLR on maximal strength, strength power, and jump and sprint performance. Subgroup analyses examined the influence of age (<20 years old, 20–45 years old, and >45 years old), gender, training status (resistance-trained and non-resistance-trained), training duration (<8 weeks and ≥8 weeks), training frequency (≤2 days/week and >2 days/week), cuff pressure (<120 mmHg, 120–180 mmHg, and >180 mmHg), and cuff width (<8 cm, 8–12 cm, and ≥12 cm) on training outcomes response. Subgroup analyses were conducted if three or more relatively homogeneous studies were available for each subgroup. The threshold for statistical significance was set at p < 0.05.

Results

Study selection

Following the initial search process, 2,925 studies were found. After removing duplicates, 1,067 studies remained for the title and abstract screening. This screening excluded 954 studies, resulting in 113 studies for full-text review. During the full-text review, 75 studies were excluded. Additionally, three more studies were identified through the references of articles. Ultimately, 41 studies were included in the meta-analysis. The search process is shown in Fig. 1. The characteristics of the studies are presented in Table 1.

Methodological quality assessment and risk of bias

Among the included studies, 12 were classified as moderate quality (scoring 4–5 points), and 28 were classified as high quality (scoring 6–10 points). With a median score of 6 out of 10, with an interquartile range of 5–7, these findings suggest that the overall quality of the studies was moderate to high, ensuring their reliability. Additionally, the GRADE assessment of the four meta-analyses determined the overall quality of evidence to be moderate. The specific PEDro scale scores and GRADE evidence profile are detailed in Supplementary Material 3. Egger’s test showed no significant risk of bias for maximum strength (b = −0.62, t = −0.63, p = 0.53) and jump performance (b = −0.83, t = −0.62, p = 0.54). Additionally, the funnel plots from the four meta-analyses revealed a fairly uniform distribution, suggesting no significant publication bias or selective reporting (see Supplementary Material 4).

Primary meta‑analysis results

The meta-analysis on maximum strength included 39 studies, with a total of 51 treatment outcome measures involving 804 participants. The effect size was −0.19 (95% Confidence Interval, CI [−0.31 to −0.06]), p = 0.003 (see Fig. 2). For muscle power, eight studies were included, with nine outcome measures involving 175 participants. The effect size was −0.04 (95% CI [−0.33 to 0.24]), p = 0.76 (see Fig. 3). In terms of jump performance, 10 studies were included, with a total of 18 outcome measures involving 187 participants. The effect size was −0.08 (95% CI [−0.30 to 0.15]), p = 0.50 (see Fig. 4). For sprint performance, six studies were included, with six outcome measures involving 110 participants. The effect size was −0.28 (95% CI [−0.71 to 0.15]), p = 0.20 (see Fig. 5).

Figure 2 Forest plot demonstrating the effects of LL-BFR training vs HLR training on maximal strength.

Effect size > 0: LL-BFR shows greater improvements than HLR. See Table 1 for studies.

Figure 3 Forest plot demonstrating the effects of LL-BFR training vs HLR training on muscle power.

Effect size > 0: LL-BFR shows greater improvements than HLR. Studies: Bradley et al. (2023), Castilla-López & Romero-Franco (2023), Davids et al. (2021), Jones et al. (2023), Tang & Qu (2022), Wang et al. (2019, 2022), Xie (2023).

Figure 4 Forest plot demonstrating the effects of LL-BFR training vs HLR training on jump performance.

Effect size > 0: LL-BFR shows greater improvements than HLR. Studies: Castilla-López & Romero-Franco (2023), Davids et al. (2021), Guang (2021), Kriley (2014), Shao (2023), Wang et al. (2019, 2022), Xie (2023), Yang et al. (2022), Zhang, Ye & Wu (2022).

Figure 5 Forest plot demonstrating the effects of LL-BFR training vs HLR training on sprint speed.

Effect size > 0: LL-BFR shows greater improvements than HLR. Studies: Castilla-López & Romero-Franco (2023), Guang (2021), Li (2020), Shao (2023), Wang et al. (2019), Xie (2023).

Subgroup analyses

Subgroup analyses were conducted if three or more relatively homogeneous studies were available for each subgroup. A total of 15 subgroup analyses were performed for maximal strength (age, gender, training status, training duration, training frequency, cuff pressure, and cuff width), muscle power (gender, training duration, and training frequency), and jump performance (age, gender, training duration, training frequency, and cuff pressure). The results showed that age, gender, training status, training duration, training frequency, cuff pressure, and cuff width did not significantly moderate the training effects (Table 2).

Table 2 Moderation analysis of individual and training factors on the effects of LL-BFR vs HLR on maximal strength, muscle power, and jump performance.

Covariate	k	ES	95% CI	I2 (%)	p-value	
Maximal strength	
Gender					0.96	
Male	29	−0.21	[−0.41 to −0.01]	35.2		
Female	8	−0.17	[−0.52 to 0.17]	0		
Mixed	14	−0.16	[−0.39 to 0.07]	0		
Age					0.29	
<20	5	0.15	[−0.30 to 0.60]	0		
20–45	34	−0.23	[−0.37 to −0.08]	0		
>45	12	−0.19	[−0.52 to 0.14]	31.5		
Training status					0.07	
Untrained	38	−0.26	[−0.40 to −0.11]	13.3		
Trained	13	0.02	[−0.23 to 0.27]	0		
Cuff pressure					0.89	
<120 mmHg	7	−0.15	[−0.48 to 0.19]	0		
120–180 mmHg	10	−0.27	[−0.66 to 0.13]	49.5		
>180 mmHg	9	−0.15	[−0.61 to 0.30]	448.2		
Cuff width					0.63	
<8 cm	18	−0.11	[−0.39 to 0.18]	44		
8–12 cm	9	−0.30	[−0.61 to 0.01]	0		
≥12 cm	12	−0.25	[−0.48 to −0.02]	0		
Duration					0.40	
<8 weeks	29	−0.24	[−0.42 to −0.06]	18.7		
≥8 weeks	22	−0.13	[−0.32 to 0.06]	0		
Frequency					0.06	
≤2 days/week	21	−0.05	[−0.24 to 0.14]	0		
>2 days/week	30	−0.30	[−0.48 to −0.12]	12.3		
Muscle power	
Gender					0.43	
Male	5	0.08	[−0.35 to 0.51]	0		
Mixed	4	−0.15	[−0.54 to 0.24]	0		
Duration					0.15	
<8 weeks	5	0.13	[−0.24 to 0.50]	0		
≥8 weeks	4	−0.32	[−0.78 to 0.15]	0		
Frequency					0.41	
≤2 days/week	4	0.08	[−0.34 to 0.50]	0		
>2 days/week	5	−0.16	[−0.57 to 0.24]	0		
Jump performance	
Gender					0.28	
Male	13	0.00	[−0.26 to 0.26]	0		
Mixed	5	−0.28	[−0.71 to 0.15]	0		
Age					0.33	
<20	6	0.09	[−0.31 to 0.50]	0		
20–45	12	−0.15	[−0.41 to 0.12]	0		
Cuff pressure					0.50	
120–180 mmHg	6	0.07	[−0.32 to 0.46]	0		
>180 mmHg	3	−0.16	[−0.71 to 0.38]	0		
Duration						
<8 weeks	8	0.10	[−0.21 to 0.42]	0	0.11	
≥8 weeks	10	−0.26	[−0.58 to 0.05]	0		
Frequency					0.67	
≤2 days/week	6	−0.14	[−0.52 to 0.23]	0		
>2 days/week	12	−0.04	[−0.31 to 0.23]	0		
Note:

Subgroup analyses were not conducted when fewer than three relatively homogeneous studies were available for each subgroup.

Discussion

The present meta-analysis compared the effects of LL-BFR training (20–30% 1RM) and conventional HLR training (60–90% 1RM) on muscle strength, power, and speed. The main findings indicated that, despite significant differences in maximal strength gains between LL-BFR and HLR training, the effect size was trivial (ES = −0.19), suggesting that both methods have similar practical effects on maximal strength. Both LL-BFR and HLR training also promoted comparable gains in muscle power and speed. Furthermore, individual characteristics (i.e., age, gender, training status) and training parameters (i.e., training duration, frequency, cuff pressure, and cuff width) did not influence the comparative training gains between the two methods. Based on the PEDro scores and GRADE assessment, the overall quality of the included studies was rated as moderate. These findings highlight the potential of LL-BFR training as a viable and effective low-load alternative to HLR training for improving neuromuscular function in healthy populations from young to older adults.

The present meta-analysis found that LL-BFR and HLR training produced similar gains in maximal strength, consistent with the meta-analysis by Grønfeldt et al. (2020) but contrasting with two other meta-analyses by Lixandrão et al. (2018) and Hughes et al. (2017). The disparate conclusions are likely due to variations in study inclusion criteria, leading to differences in study methods and sample characteristics, ultimately resulting in differing outcomes. The meta-analysis by Lixandrão et al. (2018) included studies with within-subject controls (i.e., right leg with HLR training and left leg with BFR training, potentially leading to crossover training effects) and non-randomized designs, and did not directly compare LL-BFR and HLR. Similarly, the meta-analysis by Hughes et al. (2017) included non-randomized studies and populations with osteoarthritis. In contrast, the present meta-analysis and that by Grønfeldt et al. (2020) focused solely on healthy populations and excluded studies with within-subject controls or non-randomized designs, which was expected to enhance methodological robustness. Notably, the present meta-analysis and that by Grønfeldt et al. (2020) showed higher effect sizes compared to Hughes et al. (2017) (−0.19 to −0.17 vs −0.63).

Besides the similar gains in maximal strength, our meta-analysis also found comparable improvements in muscle power and speed between LL-BFR training and HLR training. The unique effectiveness of this low-load resistance training method in enhancing neuromuscular function makes it particularly attractive in settings where high-load exercise may be contraindicated, such as basic functional training to prevent falls in frail elderly individuals, rehabilitation exercises for athletes recovering from injuries, or training for beginners. For athletes in rehabilitation, LL-BFR may be an effective method for quickly restoring competitive performance, as it not only improves maximal strength similarly to high-load resistance training but also enhances power and speed to the same extent. Additionally, even very low loading intensities (e.g., walking training) combined with blood flow restriction resulted in substantial improvements in muscle function (Ozaki et al., 2011; Abe et al., 2010). Therefore, LL-BFR training could potentially expand the options for personalized training and rehabilitation programs, particularly in situations where high-load intensities are not appropriate.

Although LL-BFR training produces less mechanical tension compared to traditional HLR training, the metabolic stress generated by blood flow restriction can compensate for this disadvantage. Studies have shown that metabolic stress is equally important as mechanical tension in promoting neuromuscular function (Moritani et al., 1992; Duchateau et al., 2021). Blood flow restriction leads to local hypoxia, increasing lactate production, which subsequently promotes the secretion of hormones (e.g., growth hormone, insulin-like growth factor-1, and vascular endothelial growth factor), thereby stimulating muscle protein synthesis (Manini et al., 2012; Yinghao et al., 2021; Ferguson et al., 2018). In a hypoxic environment, early recruitment of fast-twitch fibers occurs, and the activation of high-threshold motor units enhances the involvement of these fibers, thereby increasing muscle strength, power, and speed (May et al., 2022; Yasuda et al., 2011). Additionally, metabolic stress induces neuromuscular adaptations that increase motor unit activation, further improving muscle strength and endurance (Castilla-López & Romero-Franco, 2023; Moritani et al., 1992; Duchateau et al., 2021). Some evidence suggests that LL-BFR training may be less effective in neuromuscular activation compared to HLR training on surface electromyography (sEMG) parameters (Manini & Clark, 2009; Cook, Murphy & Labarbera, 2013). This could explain the minor differences in training benefits between LL-BFR and HLR training (trivial effect, ES = −0.19). However, higher sEMG amplitudes do not necessarily indicate greater motor unit recruitment. Therefore, more detailed and comprehensive research methods are required to accurately assess the impact of LL-BFR and HLRT on neuromuscular activation. In summary, despite the lower mechanical tension in LL-BFR training, the metabolic stress-induced physiological and neural adaptations compensate for this limitation. This makes LL-BFR training an effective training method with benefits in strength, power, and speed comparable to HLR training, particularly suitable for older adults and individuals undergoing rehabilitation.

This meta-analysis of moderator variables revealed no significant difference in training effects between durations of <8 weeks and ≥8 weeks. This underscores the advantages of LL-BFR training in terms of time efficiency and adaptability. While 3–4 week regimens in the present review can achieve results comparable to HLR training, durations longer than 4 weeks are recommended for sustained improvements. Additionally, training frequency did not have a significant moderating effect on training outcomes. However, based on practical considerations and the balance of the training load, it is recommended to perform moderate-frequency training 2 to 3 times per week. This helps to better integrate into regular training routines while avoiding overtraining or undertraining.

The relative safety of LL-BFR training has not been thoroughly investigated; however compared to other strength training methods, its safety concerns primarily stem from potential risks induced by blood flow restriction, such as blood coagulation, cardiovascular responses, and oxidative stress (Loenneke et al., 2011). It is important to note that the data presented in this meta-analysis is based on healthy participants, and the safety profile may differ for individuals with other comorbidities or disease states. This meta-analysis examined the moderating effects of occlusion pressure (70–220 mmHg or 40–80% arterial occlusion pressure, AOP) and cuff width (5–18 cm) on training outcomes and found no significant impact, which is consistent with a previous meta-analysis (Lixandrão et al., 2018). Therefore, to ensure safety while maintaining training effectiveness, it is advisable to use lower occlusion pressure. The studies included in this review utilized varied pressure prescriptions, with most employing fixed occlusion pressure values and a few using AOP percentages. Given individual differences, it is recommended to adopt personalized occlusion pressure prescriptions (i.e., AOP percentages) and account for cuff width to prevent underestimation or overestimation of occlusion pressure, which could lead to suboptimal training effects or safety issues.

Limitations

Several potential limitations of this meta-analysis warrant cautious interpretation of the findings. Firstly, the number of studies comparing the effects of BFR and HL-RT training on muscle power and speed is insufficient. This necessitates cautious interpretation of the comparative results of these two training methods and also limits our analysis and understanding of related moderator variables. Future research should focus more on comparative studies in the areas of muscle power and speed to provide more comprehensive insights. Secondly, the included studies did not report adverse reactions or injuries related to LL-BFR training. However, this does not mean that BFR training is without potential safety issues, which may be due to limitations in study design or reporting. Therefore, LL-BFR training should be applied with caution, with comprehensive risk assessment and monitoring. Thirdly, the studies included varied in their occlusion pressure prescriptions, adding complexity to the analysis of moderator variables. Since individual responses to the same pressure can differ, personalized blood flow restriction protocols are crucial. Future BFR training research should adopt occlusion pressure prescriptions based on arterial occlusion pressure (AOP percentages) to better accommodate individual differences and ensure the effectiveness and safety of the training.

Conclusions

The studies included in this meta-analysis were of moderate quality. This analysis indicates that LL-BFR and HLR training produce similar effects on muscle power, jump performance, and speed, with LL-BFR showing slightly less improvement in maximal strength compared to HLR. Additionally, individual characteristics (i.e., age, gender, and training status) and training parameters (i.e., training duration, frequency, cuff pressure, and cuff width) do not significantly moderate the training effects of either method. Therefore, LL-BFR training appears to provide a viable and effective alternative to traditional HLR training, suitable for healthy individuals of all ages.

Supplemental Information

Supplemental Information 1 Search Alert.

Supplemental Information 2 Raw Data.

Supplemental Information 3 Means and Standard Deviations (data).

Supplemental Information 4 PRISMA checklist.

Supplemental Information 5 Funnel Plots.

Additional Information and Declarations

Competing Interests

The authors declare that they have no competing interests.

Author Contributions

Tao Wang conceived and designed the experiments, performed the experiments, analyzed the data, prepared figures and/or tables, authored or reviewed drafts of the article, and approved the final draft.

Yutong Liu conceived and designed the experiments, performed the experiments, analyzed the data, prepared figures and/or tables, authored or reviewed drafts of the article, and approved the final draft.

Xiaolin Wang performed the experiments, analyzed the data, prepared figures and/or tables, and approved the final draft.

Saidon Amri performed the experiments, authored or reviewed drafts of the article, and approved the final draft.

Tengku Fadilah Kamalden performed the experiments, analyzed the data, authored or reviewed drafts of the article, and approved the final draft.

Zhendong Gao performed the experiments, prepared figures and/or tables, and approved the final draft.

Yee Guan Ng conceived and designed the experiments, performed the experiments, authored or reviewed drafts of the article, and approved the final draft.

Data Availability

The following information was supplied regarding data availability:

The raw measurements are available in the supplemental Material 2.

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
