# Peer review of "Effect of blood-flow restricted vs heavy-load resistance training on strength, power, and speed for healthy volunteers: a systematic review and meta-analysis"

_PeerJ, doi:10.7717/peerj.19110_

## Round 0.1 · original submission · Major Revisions

We thank the authors for submitting their interesting manuscript. Two independent reviewers reviewed the manuscript and raised substantial concerns which need to be addressed before a potential acceptance decision can be made.

Reviewer 1 ·

Basic reporting

Title & Abstract
Title
The title is clear but acknowledgement that this systematic review only included non-pathological individuals is important. I recommend amending the title to “Effect of Blood-Flow Restricted Vs Heavy-Load Resistance Training on Strength, Power and Speed for Healthy Volunteers: A Systematic Review and Meta-Analysis” or something to that effect.
Abstract
Method:
• PDF Page 5, Line 34: I recommend the authors state the search dates, in accordance with the PRISMA recommendation.
• Page 5, Line 37-38: The sentence “In cases where data were unavailable, they were requested directly from the authors” – I recommend this be removed as this is not critical information for the Abstract and the abstract word count can be used in the following...
• I recommend the authors state the critical appraisal tool used i.e. PEDro tool. This would improve reporting.
• Provide one sentence outlining the data analysis methods adopted. This would improve reporting to aid interpretation in the following section.
Results:
• PDF Page 6, Line 40-41: “enhancing maximal strength (ES = -0.19, 95% confidence interval [CI]: [-0.31, -0.06], p < 0.01)” – this should be reconsidered as the evidence suggests a difference between the training approaches for maximal strengthen. Whilst I agree in not difference (clinically or statistically) for the other outcomes, some may argue that this outcome is different. I recommend the authors reconsider this interpretation and revise the text accordingly.
• PDF Page 6, Line 39: There is no reporting of the critical appraisal results which are a prerequisite for a systematic review. I recommend a statement is made in the Abstract Results reporting the PEDro scale finding to report this OR synthesize the outcomes using the GRADE approach so the reader can understand the certainty or quality of evidence underpinning these analyses.
Conclusions:
• PDF Page 6, Line 47: I agree with the concluding statement but recommend the authors also mention this to be the finding for ‘healthy’ or ‘non-pathological’ individuals as blood-flow restriction therapy is being used increasingly for clinical populations. As stated above, I also recommend they reconsider their ‘blanket’ statement of comparability as the data suggests a between group difference for maximal strength.

Introduction
This is comprehensive, clear and well-articulate. There is a clear rationale for performing the study. I have limited information to add to the current Introduction.
PDF Page 6, Line 72: The authors may want to reflect on some evidence in pathological groups which help explain the context of this research question. This included referring/considering studies such as:
• Ahmed I, Mustafaoglu R, Erhan B. The effects of low-intensity resistance training with blood flow restriction versus traditional resistance exercise on lower extremity muscle strength and motor function in ischemic stroke survivors: a randomized controlled trial. Top Stroke Rehabil. 2024 May;31(4):418-429. https://www.tandfonline.com/doi/abs/10.1080/10749357.2023.2259170
• Wen Z, Zhu J, Wu X, Zheng B, Zhao L, Luo X, Wu Z. Effect of Low-Load Blood Flow Restriction Training on Patients With Functional Ankle Instability: A Randomized Controlled Trial. J Sport Rehabil. 2023 Aug 9;32(8):863-872. https://journals.humankinetics.com/view/journals/jsr/32/8/article-p863.xml
• Nitzsche N, Stäuber A, Tiede S, Schulz H. The effectiveness of blood-flow restricted resistance training in the musculoskeletal rehabilitation of patients with lower limb disorders: A systematic review and meta-analysis. Clin Rehabil. 2021 Sep;35(9):1221-1234. The effectiveness of blood-flow restricted resistance training in the musculoskeletal rehabilitation of patients with lower limb disorders: A systematic review and meta-analysis.
As well as (from a timing, dose and load perspective with healthy participants):
• Fabero-Garrido R, Gragera-Vela M, Del Corral T, Izquierdo-García J, Plaza-Manzano G, López-de-Uralde-Villanueva I. Effects of Low-Load Blood Flow Restriction Resistance Training on Muscle Strength and Hypertrophy Compared with Traditional Resistance Training in Healthy Adults Older Than 60 Years: Systematic Review and Meta-Analysis. J Clin Med. 2022 Dec 13;11(24):7389. https://www.mdpi.com/2077-0383/11/24/7389
• Bechan Vergara I, Puig-Diví A, Amestoy Alonso B, Milà-Villarroel R. Effects of low-load blood flow restriction training in healthy adult tendons: A systematic review and meta-analysis. J Bodyw Mov Ther. 2024 Jul;39:13-23. https://www.sciencedirect.com/science/article/pii/S1360859223002565
Overall, the article is clear, unambiguous and well-presented grammatically. The terminology and phraseology is appropriate for an academic paper.

Figures & Tables
I have no concerns with regards to image integrity and manipulation but recommend the authors supply better resolution images for the PRISMA flow-chart (Figure 1) and forest plots (Figure 2-5) to improve presentation.
I was unable to assess Supplementary Files as these were not presented for examination. However, this did not significantly impact on my abilities to review this paper. Where they are mentioned, these appear appropriately placed and of appropriate content for a supplementary file attachment.

Experimental design

Material and Methods
The search (PDF Page 7, Line 100), eligibility assessment (PDF Page 7, Line 112) and critical appraisal (PDF Page 8, Line 120) were appropriately conducted and reported. Please explain your decision or reasoning
PDF 8, Line 130: The authors should consider stating what the primary outcome and primary endpoint was. The team have needed to complete this or reflect on this for the PROSPERO registration and therefore I recommend that reconsider this and make a statement of this, dividing primary and secondary outcomes for this review.
PDF 8, Line 137: I recommend the authors reconsider the sentence “Notably, when multiple time points were assessed for outcome variables, the data from the last time point was considered the post-training value for analysis.” – if there are multiple time-points, it would make more sense to pre-specify them so there is not a large outcome in post-training timescale i.e. post-training could theoretically be post-exercise or 7 days depending on the outcome assessment point for an individual study. Defining these as ‘immediate’, ‘short-term’ and ‘long-term’ (for example) with immediate being post-exercise, short-term being post-exercise to 3 days, and longer-term being four days onwards (for example) would be more robust and reduce the risk that time-point variation impacts on analysis outcomes. I recommend the authors consider this and reflect such a change in the revisions.
Page 8, Line 140: The authors are recommended to state how study heterogeneity was assessed. The current sentence “Considering measurement variability and heterogeneity among the included studies, a random effects model was applied” needs more clarity. Was this by visual assessment of the data extraction table (for example). I recommend the authors explain this in the text, particularly as variation between studies were evident in frequency, dose and type of exercise performed (for example). Nonetheless, I agree with the author’s approach of assessing all outcomes using a random-effects model to account for this potential ‘known’ variability between studies included in the meta-analyses.
Page 9, Line 160: I recommend the authors mention their methods of assessing publication bias. This is reported in the Results (Page 10, Line 189-193) but needs to be mentioned in the Methods as well for comprehensive reporting.
Page 9, Line 160: The authors may also consider assessing each of their principal 4 meta-analyses using the GRADE approach. This would then synthesize both the meta-analysis results with the critical appraisal, publication bias, statistical heterogeneity and application/generalizability elements which may be a methodological advantage to further strengthen this paper. I recommend the authors consider this suggestion in their reporting. This should then be added ot both the Methods and reported in the Abstract and Results sections (at the least).

Validity of the findings

Results
These are largely clearly reported. The results expected to be presented in this section reflect those which are needed to answer the research question and derived from the methods reported in the paper’s Methods section. The evidence is both robust and plausible. The evidence is also valuable in advancing current knowledge in the field. I have identified two key points the authors should consider in a revision.
Results, Page 9, Line 177: As well as presenting the median score, I recommend the authors also provide the inter-quartile ranges to gain an appreciation on the precision of this appraisal result. This would improve reporting.
Page 9 Line 197- Page 10, Line 203: The authors should subdivide the presentation of meta-analysis using subheadings to Principal Meta-analysis and then Subgroup Analyses. Whilst the principal meta-analysis results are clearly communicated with data, there is a need to provide more data within the subgroup analyses rather than expecting the reader to cross-reference to a Supplementary File. I recommend the authors incorporate and present more data within the text of the subgroup analyses as these are one reason why this paper may offer new and novel findings over other systematic reviews in the field.

Discussion
This section is well-communicated with appropriate interpretation of the meta-analysis findings with previous literature and supporting evidence which is sufficiently robust and recently published. The limitations section is appropriate and well-judged. I have identified a number of points to further strengthen this interpretation.
Page 10, Line 214: I recommend the authors provide a statement on the critical appraisal results. This is important to ensure that the reader is aware the evidence is moderate in quality based on the appraisal assessment.
Page 10, Line 227: the authors are recommended to provide an estimated impact of the statement “focused solely on healthy populations and excluded studies with within-subject controls or non-randomized designs” - I recommend they add whether they think the effect size is enhanced or diminished as a result of their methods compared to these other meta-analyses. This would put their findings into greater context when trying to compare them against previous literature.
Page 10, Line 230: the sentence starting “The unique effectiveness of this low-load resistance training method in enhancing neuromuscular function makes it….” feels like a separate point and therefore I recommend the authors consider presenting this in a separate paragraph.
Page 11, Line 226: the sentence “. Even a 3-week LL-BFR training regimen can yield benefits comparable to HLR training” does not necessarily fit in this text and pointing out this specific three-week programme over other studies feels to be at risk of a little ‘selection bias’ in reporting. I recommend the authors reconsider whether presenting this sentence is essential in supporting this point.
Page 11, Line 273: The authors should emphasize in this paragraph that all the data presented were based on ‘healthy participants’ and that safety profile may be different for those with other comorbidities or disease-states. This is an important acknowledgement to make somewhere in this paragraph.

Conclusion
The summary conclusion reflects the results presented. Please explain your decision or reasoning except maximal strength was reported as different between the groups and should be reported as such in Page 12, Line 305.
Page 12, Line 304-305: There should be some acknowledgement that the evidence is moderate quality. I recommend this is made explicit in the opening sentence of this section, summing up the results.

Additional comments

no comment

·

Basic reporting

This manuscript is well written, provides adequate background, is structured appropriately, and reports results that are congruent with the stated aims.

Experimental design

This is a meta-analysis.

Validity of the findings

Based on the reported methodology, I believe the results reported are appropriate and valid.

Additional comments

General comments
Line 58 – How is high load training “clearly unsuitable” for these populations when high-load training is defined as a relative percent to someone’s 1RM? This should be rephrased to better indicate what is meant.
Line 69 – What adaptations specifically in the referenced papers?
Line 73-75 – In what populations?
Line 124 – Repeated word.
Line 141 – Which specific R packages and versions were used?
Table 1 – Missing units (y) on some ages in the Subject column
Table 1 – Does sample size (N) indicated as x/y indicate the number of participants in LL-BFR (x) and HLR (y)? Please specify.
Table 1 – Is there a way to specify number of men vs. women in studies that included both genders?
Figures 2-5 – I would recommend you change “experimental” and “control” to “LL-BFR” and “HL” to align with terms used throughout the manuscript. On the forest plots, which direction from 0 indicates more favorable improvements for LL-BFR or HLR? Please indicate this.

I believe you define “training status” throughout as whether the participants are athletes. In Supplementary Table 5, you call this “sports status.” I would select one of these terms, define it, and be consistent throughout the manuscript. My concern is that just because an individual plays a sport, it does not mean they are necessarily resistance trained.

Line 245 and 252 – The review article cited here (reference 12) is specific to the impacts of mechanical and metabolic stress on muscle hypertrophy, not neuromuscular function, as mentioned.

I would consider moving Supplementary Table 5 into the manuscript as Table 2 as relevant and important results based on moderation analysis are presented.

---

## Round 0.2 · Minor Revisions

After second revision their still remain minor concerns, which need to be addressed before publication. The main focus should lay on strengthening the methodological approach.

Reviewer 1 ·

Basic reporting

Title & Abstract
Title: This is now clear following the recommended correction.
Abstract: All recommended changes have been made. This is now clearly reported. I have no further recommendations.

Introduction
This is now clear and well reported. The addition of the recommended citations has further strengthened the rationale for the study. I have no further recommendations.

Experimental design

Material and Methods
The authors have improved the Methods section. However, there remain a small number of important points to address which have not been comprehensively completed following the initial review.
Methods: Line 120: Page 4: Registration and Literature Review: The statement “After removing duplicates, the screening process was conducted in three phases: titles, abstracts, and full texts, following PRISMA guidelines” should be reconsidered. PRISMA reporting checklists do not offer recommendations on methods. PRISMA is a reporting checklist. I therefore recommend the authors either find a methodological citation to support this, such as the Cochrane Handbook for Intervention Systematic Reviews, or remove the ‘following PRISMA guidelines’ statement at the end of this sentence.
Methods: Line 154: Page 4: Data Extraction: Justification on time-point was acceptable although this has still not addressed the point on whether there is variability in analysis by outcome i.e. post-treatment data may have been pooled with 6 weeks post-training data. This is a source of methodological heterogeneity. I recommend the authors reconsider this recommended amendment to ensure that the meta-analysis was not undermined by differences in data time-point.
Methods: Line 180: Page 5: Statistical Analysis: The authors have outlined how they assessed statistical heterogeneity, but they still have not stated how methodological or study heterogeneity was assessed. This is crucial to provide robustness to the analysis, particularly reflecting on the point on time-point variability above. I recommend the authors state how methodological heterogeneity was determined i.e. was this by observation of the data extraction table for study design, population, intervention and assessment?

Results
All points raised in the original review from myself and Reviewer 2 have been addressed. This section has been appropriately addressed. The addition of the GRADE assessment is a positive. I have nothing further to recommend.

Validity of the findings

Discussion
The points raised by myself and Reviewer 2 are addressed. The concerns regarding interpretation and limited acknowledgement of the certainty of evidence have been addressed. This section is stronger and I have nothing further to recommend.

Conclusion
The conclusion is clear and the addition of acknowledgement of certainty of evidence is appropriate. I have nothing further to recommend in this section.

Figures & Tables
These are clear and appropriately presented. There are no image integrity or manipulation issues detected.

Please refer to our editorial polices for guidelines on image integrity and manipulation. If you have any concerns about duplication or manipulation of images, please provide detail in the box above

Additional comments

NA

---

## Round 0.3 · Minor Revisions

We thank the authors for carefully providing responses to the reviewers comments. While reviewer 2 is happy with accepting the manuscript, there are some minor comments from reviewer 1, which we would like you to address.

Reviewer 1 ·

Basic reporting

Title & Abstract
Title: This is clear and appropriately reported. I have nothing further to recommend.
Abstract, Page 6, Line 30 to Page 7, Line 54: This is sufficiently clear. I have nothing further to report or recommend.

Introduction
Introduction, Page 7, Line 58 to Page 8, Line 101: This is clearly communicated. The rationale is clear and based on recent literature. The research question is stated. I have nothing further to recommend.

Experimental design

Material and Methods
Methods, Page 8, Line 103 to Page 10, Line 183: The authors have addressed the points raised in the last review. I have one minor point to suggest to improve the interpretation following the authors change.
Methods, Statistical Analysis, Page 10, Lines 175-177: The sentence “Methodological heterogeneity, including differences in study design, participant characteristics, and intervention protocols, was systematically assessed and accounted for through subgroup analyses” is an important addition. To aid reporting, I recommend this is moved to Line 167 so the full explanation of methodological and statistical heterogeneity is presented together. I recommend the text is therefore:
“…Methodological heterogeneity, including differences in study design, participant characteristics, and intervention protocols, was systematically assessed and accounted for through subgroup analyses. Statistical heterogeneity was assessed by evaluating the variation in effect sizes across studies, the overlap of confidence intervals, as well as quantitatively using the I² statistic. I 2 values < 25% indicating low heterogeneity, 25-75% indicating moderate heterogeneity, and > 75% indicating high heterogeneity [36]…”
This would provide a clearer interpretation of how heterogeneity assessment was considered in the analysis.

Validity of the findings

Results
Results, Page 19, Line 185 to Page 11, Line 118: The results are clearly presented. The interpretation is appropriate. The acknowledgement and interpretation of the end-of-intervention timepoint is now clearer and interpretation is appropriate, respecting this. The Figures and Tables appropriately support the data (Table 1-2; Figure 1-5; Supplementary Materials 3 and 4). Supplementary Materials 3 illustrates the GRADE assessment. These are appropriately interpreted. I have nothing further to recommend.

Discussion
Discussion, Page 11, Line 232 to Page 14, Line 335: This is appropriate. The Discussion has been grounded in appropriate literature. The interpretations from the results are reasonable and clearly communicated. I have nothing further to recommend.

Additional comments

Conclusion
Conclusion, Page 14, Line 337 to 345: This is appropriate. The interpretation from the meta-analysis is clearly made and the results are not overstated. I have nothing further to recommend.

Figures & Tables
The Figures and Tables are clear and legible. There are no unnecessary modifications and I did not detect any issues regarding image integrity or manipulation.

·

Basic reporting

The manuscript is written very clearly, and all methods and results are presented well. Sufficient background is provided in the introduction.

Experimental design

The methodology of the meta-analysis is appropriate and carried out well.

Validity of the findings

The findings and results presented here are in accordance with previous research. The authors explain how their results differ from previous similar meta-analyses on the topic based on their methodology and inclusion criteria.

Additional comments

This manuscript has been greatly improved following the revisions. The Methods section is much clearer, and the results are presented well. The authors have done a good job presenting their results in the context of other recent meta-analyses as well as discussing the limitations of their study and BFR in general.

---

## Round 0.4 · accepted · Accept

I congratulate the authors on a well improved manuscript.